# Effect of the Modifications on the Physicochemical and Biological Properties of β-Glucan—A Critical Review

**DOI:** 10.3390/molecules25010057

**Published:** 2019-12-23

**Authors:** Hongjie Yuan, Ping Lan, Yan He, Chengliang Li, Xia Ma

**Affiliations:** 1School of Perfume and Aroma Technology, Shanghai Institute of Technology, Shanghai 201418, China; yhj15216685820@126.com (H.Y.); yvonnehe2004@126.com (Y.H.); 2Guangxi Key Laboratory of Polysaccharide Materials and Modification, School of Chemistry and Chemical Engineering, Guangxi University for Nationalities, Nanning 530008, China; gxlanping@163.com; 3LB Cosmeceutical Technology Co., Ltd., Shanghai 201499, China; Richard-li@laibo.com.cn

**Keywords:** polysaccharides, β-glucan, soluble, biological activity, modification, anti-oxidation, functions

## Abstract

β-Glucan exhibits many biological activities and functions such as stimulation of the immune system and anti-inflammatory, anti-microbial, anti-infective, anti-viral, anti-tumor, anti-oxidant, anti-coagulant, cholesterol-lowering, radio protective, and wound healing effects. It has a wide variety of uses in pharmaceutical, cosmetic, and chemical industries as well as in food processing units. However, due to its dense triple helix structure, formed by the interaction of polyhydroxy groups in the β-d-glucan molecule, it features poor solubility, which not only constrains its applications, but also inhibits its physiological function in vivo. One aim is to expand the applications for modified β-glucan with potential to prevent disease, various therapeutic purposes and as health-improving ingredients in functional foods and cosmetics. This review introduces the major modification methods required to understand the bioactivity of β-glucan and critically provides a literature survey on the structural features of this molecule and reported biological activity. We also discuss a new method to create novel opportunities to exploit maximally various properties of β-glucan, namely ultrasound-assisted enzymatic modification.

## 1. Introduction

Ever since β-glucan was first found to possess bioactivity and used in clinical practice, many scholars have devoted research to β-glucan from different sources, such as bacteria, fungi and higher plants; the cell wall of *Saccharomyces cerevisiae* is also an important source of β-glucan [1]. The cell wall of yeast is divided into three layers from outside to the inside, namely mannan, protein and dextran; β-d-glucan is the main component of dextran (Figure 1) [2,3]. Yeast β-d-glucan accounts for 30–60% of yeast cell wall dry weight. About 60% of yeast β-d-glucan is a long chain linked by 1500 glucose residues via β-(1-3)-glycoside bonds and its molecular weight is about 240 kDa [4,5,6]. A large number of studies have shown that yeast β-d-glucan can activate macrophages to initiate anti-tumor [7], anti-bacterial [8], wound healing [9], antioxidant [10,11], and lipid-lowering [12] effects. In addition, yeast β-d-glucan is widely used in the food industry as a thickener, emulsifying stabilizer and fat substitute, because of its good water-holding, heat preservation, film-forming, and non-irritating properties [13,14,15,16]. In recent decades, there has been an increasing interest in the utilization and study of β-glucan, particularly the establishment of structure–function relationships, for various novel applications owing to their biocompatibility, biodegradability, and non-toxicity, and also for some specific therapeutic activities. 

Yeast β-glucan (Figure 2) belongs to the class of β-glucan, and its structure includes two distinct macromolecular components comprised of consecutively (1→3)-linked β-d-glucopyranosyl residues, with small numbers of (1→6)-linked branches and a minor component with consecutive (1→6)-linkages and (1→3)-branches (Figure 2a). β-d-glucan can be found as random coils or more organized conformations that constitute a network comprised of single helix chains associated as triple helices that are stabilized by inter- and intra-hydrogen bonds. The intramolecular polyhydroxy groups interact to form a dense triple helix structure resulting in insolubility in water and organic solvents, such as ethanol (Figure 3) [6,12,17]. Moreover, high molecular weight branched β-glucans from yeast and fungal cell walls, which contain variable but minor proportions of (1→6)-β-glucosidic interchain linkages, are insoluble in water. Because of their insoluble chemical nature, particulate β-glucans are not suitable for many medical and food applications. Moreover, the structure is unfavorable for penetrating multiple cell membrane barriers into organisms to exert pharmacological effects, mainly because the molecular weight of yeast glucan is too high. Conversely, polysaccharides will not show same bioactivity if the molecular weights are too low [18,19]. Many studies have revealed that the biological properties of various biopolymers depend on their molecular weight. Bohn and BeMiller [20] reported in 1995 that β-glucans molecular weights circa 4100 kDa are considered to be high molecular weight compounds, and the low molecular weight range is considered to be less than 50 kDa. Various methods of changing or modifying the β-glucan chemical structure, and transforming it to a soluble form, have been published. All active β-glucans share the same basic structures, which can be changed by modifications leading to a conformational transformation in solution that can directly affect bioactivity. Several parameters characterize the natural β-glucan: primary structure, solubility, degree of branching (DB), molecular weight, as well as the polymer charge and/or solution conformation (triple helix, single helix, or random coil conformation) (Figure 3). All these factors play a role in glucan-associated biological activity [21,22]. With a high degree of polymerization (DP > 100), β-glucans are completely insoluble in water. The solubility increases as the degree of polymerization of the β-glucan is decreased. Generally, β-d-glucan with a degree branching between 0.04 and 0.75 has biological activity. The solubility decreases with an increase in the degree of branching. The increase of the yeast glucan water solubility could transform polysaccharides into ideal injection or other dosage forms, permitting them to better exert their biological activity. Most of the molecular modifications bring about an increase in the antioxidant activity of polysaccharides, and among these, sulfated and acetylated modifications are very common [23]. Thus, it can be deduced that the bioactivity of yeast glucan and their physicochemical properties are restricted by their structure. Therefore, modifying the structure of yeast glucan is fundamental to providing more options for screening biological activity or specific polysaccharide agents. Hence, many physical, chemical, and biological methods have been applied in order to modify β-glucan to achieve the desired bioactivity. 

Natural β-glucans have ideal bioactivities, but they are not always satisfactory. It has been found that structural modifications greatly affect the biological activity of β-glucan. However, there are few papers that have reviewed the relationship between changes in the physicochemical properties and bioactivities of β-glucan modification methods. In addition, the impact on the solubility, molecular weight, solution conformation and biological activity of the modified β-glucan has rarely been summarized and reviewed. The main aims of the present review are to discuss recent advances in research on the modification and solubilization of β-glucan, its sources, structural features of natively β-glucan, and the possible mechanisms of improved bioactivity through such modifications. 

## 2. Advances in Research Yeast on β-Glucan Modification

Molecular modification can change the polysaccharide dimensional structure, molecular weight and the substituent group types, numbers, and positions, with a profound impact on bioactivity. To date, dozens of molecular modification methods have been tried including carboxymethylation, sulfation, selenylation, phosphorylation, ultrasonic disruption, and the degradation of polysaccharides, which are generally classified as chemical, physical and biological modifications (Figure 4). Molecular modifications remarkably affect the physicochemical properties of polysaccharides and changes their physical and chemical properties including solubility, relative molecular mass, and intrinsic viscosity (Table 1). Generally the modified polysaccharides display better biological properties because of changes in their physicochemical properties. 

### 2.1. Physical Modification Method

The physical method solubilizes and functionalizes the polysaccharide by breaking the main chain of the glucan macromolecule through mechanical action without destroying the basic molecular structure of yeast β-glucan. After physical treatment, lower molecular weight fractions of β-glucans can be obtained, which not only show improved water solubility and the physiological function of β-glucans, but also exhibit conformation change in solution. The operation is simple, no chemical reagent is used and the environment is not polluted. It is mainly used for the modification of polysaccharides that already have bioactivity before treatment in order to reduce their molecular weight and increase their bioactivity efficacy; it can be used to improve the function of some medical materials. However, the yield of soluble yeast glucan is low if prepared by physical methods alone. The most commonly used methods are ultrasonic disruption, irradiation-induced reactions, microwave exposure. 

#### 2.1.1. Thermal Degradation 

The thermal degradation method refers to the degradation process of the polymer when it is heated, and is an important approach to degrade polymers. The method is simple and the product is easy to obtain with uniformity. It is a solubilization method suitable for functional foods [24], and by controlling the thermal degradation cycle, products of different molecular weights can be prepared [25]. In 2009, Zhang et al. [26] studied the effects of heat treatment on the transformation of β-glucan solution of *Pleurotus ostreatus*. It was found that heat treatment of *Pleurotus ostreatus* β-glucan changed the solution conformation. The first approach was at a temperature of 8~12.5 °C, when the solution conformation changed from a polymer bundle to a small helical cluster. When the temperature was raised to 40–60° C, a second transition occurred, transforming the molecule from a helical cluster into a single polysaccharide chain. In 2018, Ishimoto et al. [27] applied the heat degradation method to solubilize yeast β-glucan and found that it was effectively solubilized by treatment in an aqueous suspension at 135 °C for several hours. Baker’s yeast cell wall β-glucan (BBG) was resuspended in water for injection, and heat-treated at 100, 121, or 135 °C. The solubilization ratio of BBG was examined after several reaction periods at three heating temperatures. As a result, the insoluble BBG first swelled at ∼1 h, and the insoluble component was reduced by continued heating; solubilization at 135 °C was much greater and the solubilized ratio increased with the heating duration. Continued heating also increased the percentage of the lower molecular weight fragments. The molecular weight distribution was clearly shifted to the lower end of the spectrum by heat treatment for 4 h or more.

#### 2.1.2. Irradiation 

Irradiation refers to a physical modification method that induces physicochemical changes of β-glucan by ionizing radiation such as gamma rays, X-rays, and electron beams [28]. The average molecular weight of irradiated β-glucan significantly decreased as the irradiation dose increased. In addition, irradiation improved the solubility and decreased the viscosity of β-glucan by the radiolysis of the glycosidic bonds, and this effect was dependent upon the absorbed dose [29,30]. Radiation can break down β-glucan and change the primary structure, so that cleaved fragments of polysaccharides can be formed. Research conducted by Methacanon et al. [31] with various doses (0–100 kGy) of γ-irradiation was shown to be a clean method to produce different low molecular weight molecules. The results showed that the average molecular weight of the irradiated molecules significantly decreased as the radiation dose increased, whereas the functional groups before and after irradiation, detected by FTIR and ^13^C-NMR, were identical. It is worth noting that the molecular weight decreased rapidly at first and subsequently leveled off to a much slower rate at dosages > 5 kGy, and subsequently the molecular weight remained almost unchanged at the oligomer level. Khan et al. [32] evaluated the changes in structure, function, antioxidant and antibacterial properties of yeast β-d-glucan after different irradiation intensities and found that γ-irradiation had a great influence on the functional properties of yeast β-d-glucan. The viscosity and swelling power of irradiated samples significantly decreased as the irradiation dose increased. In contrast, the binding strength, emulsifying properties, foaming properties and bile acid binding strength all increased. All antioxidant properties examined using six different analytical methods were significantly increased in a dose-dependent manner. As the irradiation dose increased from 5 kGy to 50 kGy, the antibacterial activity of yeast β-d-glucan also exhibited an upward trend.

#### 2.1.3. Ultrasonication 

The theory of this method is that low frequency and high-intensity ultrasonic energy can break some of the chemical bonds of pyran dextran by increasing the vibratory energy of molecular particles. It generally leads to depolymerization and side chain breakage within the molecules producing oligo-sugars, making them an effective and efficient method for polysaccharide modification. Future oligosaccharide and nano scale molecule processing will mostly probably be carried out using a sonication process [33]. Following ultrasound treatment, the molecular weight distribution curves gradually shift to the low molecular weight region with a narrower distribution and the chain degradation reaction follows a random scission model. The viscosity of modified glucan solution drops remarkably and the glucan after ultrasonication is better dissolved in water [34,35,36]. Hunter et al. [37] successfully obtained an immunologically active, homogeneous, non-aggregated, micro-particulate 1–2 µm diameter β-glucan-containing material from the budding yeast *Saccharomyces cerevisiae* by using a combination of sonication and spray-drying. This microparticulate β-glucan could remain in suspension longer than the aggregated form. Wang et al. [38] sonicated the large-sized yeast β-glucan (SCG) at a working frequency of 18 to 21 kHz at 400 W, with a processing time of 24 s per cycle and an interval of 6 s for a total of 20 cycles. The average particle size of the SCG was changed from the initial 56.49 µm to 2.33 µm, and the ultrasonic modification effect was remarkable. The modified SCG (C-SCG) showed a uniform distribution of single particles, and the particle size distribution bandwidth was narrowed. Chen et al. [39] proved that ultrasonic treatment is an effective physical method for improving the solution properties of water-insoluble homoglycans such as β-glucan without changing its primary chemical structure, and such modified polysaccharides had enhanced antitumor activity. The results showed that ultrasonic treatment could open up the compact structure of *Poria cocos* polysaccharide (PCS) into a linear chain. 

#### 2.1.4. High-Pressure Micro-Jet Homogeneization

The high-pressure micro-jet homogenizer (HPM) is a new type of nano-scale homogenization device, a physical method for transferring mechanical energy to a fluid particle under high pressure using ultrasonic radiation and mechanical shear to changes structure and control the molecular weight of polymers such as proteins [40,41]. A number of polysaccharide studies involving high-pressure micro-jets have demonstrated the ability of micro-jets to cause molecular hydrogen bond cleavage and reorganization, side chain breakage [42] and thus change the microscopic morphology [43], and even increase solubility and changes in functional properties [44,45]. Huang et al. [46] found that the molecular weight and monosaccharide composition of *Mesona chinensis* Benth polysaccharide (MP) were changed after DHPM treatment. They treated MP (1 mg/mL) using a laboratory scale dynamic high-pressure microfluidizer at 120 MPa for six cycles and dialyzed the sample treated with DHPM (MW cut-off 10 kDa) vs. distilled water for 24 h, then freeze-dried it to obtain polysaccharide DMP. Gao et al. [47] modified the solubilized yeast β-glucan with a high-pressure micro-jet combined with ionic liquid to prepare soluble β-glucan and studied the modification effects and structure of the circulating ionic liquid. The ionic liquid 1-ethyl-3-methylimidazolium acetate was combined with high-pressure microfluidic modified solubilized yeast β-glucan and treated at 175 Mpa in 200 mL of ionic liquid for 9 min, concentration 0.5%; the total sugar content of the modified dextran obtained was 97.22 ± 0.54% and the yield was 79.25%. The structure of the new and old ionic liquids was compared by FTIR and 1H NMR techniques. The results revealed that the high-pressure microfluid treatment did not destroy the structure of the ionic liquid. 

#### 2.1.5. Supercritical Fluid Technology 

Supercritical CO_2_ (SCCO2) drying technology is a new technology for preparing special materials with green safety and controllable product structure. The use of supercritical drying, to remove the solvent contained within the gels without collapsing its three dimensional polymeric network, provides unique highly porous materials known as aerogels. Biopolymer aerogels are a special class of lightweight highly porous structured materials that are of interest for their low densities, high surface areas, low heat conductivities, and mechanical strength [48,49]. Use of SCCO2 for the preparation of aerogels offers the advantage of eliminating the use of organic solvents and being able to operate at relatively low temperatures to minimize the degradation of bioactives. These aerogels are being investigated for applications such as component separation, absorbents, catalysts, and supports for chemical reactions, drug delivery and scaffolds for tissue engineering. Comin et al. [50] obtained an aerogels using SCCO2 drying with β-glucan at a 5% level and flax mucilage at a 10% level. SCCO2-dried polysaccharide aerogels showed promise for use as a delivery vehicle for nutraceuticals including flax SDG. Additionally, biobased aerogels also possess interesting functionality, chemistry, biodegradability, biocompatibility and sustainability inherent to biobased polymers. Simultaneously, Comin et al. [51] determined the effect of the β-glucan concentration and drying technique on the resulting characteristics of aerogels. A biodegradable, biocompatible, renewable and edible β-glucan aerogel formed using a barley β-glucan concentrate at 5%, 6%, and 7% (*w/v*) levels was dried using SCCO2. The results showed an increasing β-glucan concentration (5–7%) had very few effects on the aerogel properties. SCCO2 aerogels had lower densities than air-dried gels and a more consistent structure than freeze-dried gels. Supercritical carbon dioxide is also used for the extraction of β-glucan. Sibakov et al. [52] demonstrated that lipid removal with supercritical carbon dioxide enhanced the separation of oat β-glucan, starch, and protein in distinct fractions and obtained higher β-glucan content than existing products produced with dry fractionation techniques. This was probably due to more efficient milling yielding smaller particles, and the release of starchy material from cellular structures during milling of defatted oats. 

### 2.2. Chemical Modification Method

Chemical modification is the most widely used method as it can significantly increase the water solubility and bioactivity of polysaccharides by ‘grafting on’ other moieties. Although the introduction of the new group imparts some new activity to the dextran, its original structure has been changed, so its application is limited to some extent. Moreover, the treatment method can easily form residues, and some operations are also faced with a series of shortcomings such as a complicated modification process, high modification costs and the need for special modification equipment. Although chemical modification can increase the solubility of β-d-glucan, it will also alter its natural structure, changing its biological activity. Research has shown that the sulfonation of fungal exocellular β-(1→6)-d-glucan can change its original structural characteristics, thus improving its water solubility. Moreover, Williams and co-workers (1991) reported a method to modify micro-particulate β-d-glucan, in which the insoluble glucan was turned into glucan phosphate and the water solubility of the modified glucan was significantly improved. Chemical modification methods include sulfation, alkylation, carboxymethylation, phosphorylation, selenization, and acetylation. 

#### 2.2.1. Sulfation 

Sulfation modification is one of the most commonly used methods to modify polysaccharides. Specifically, sulfated polysaccharides are synthesized by substituting hydroxyl, carboxyl, or amino terminal groups with sulfate groups, with improvements of biological activities. Among them, the solubility of the modified glucan is increased thanks to the hydrophilicity of the sulfate group [53,54,55]. The good aqueous solubility of the resultant polysaccharide was a huge advance to its application especially in pharmaceuticals. A novel glucan sulphation process is proposed to improve polysaccharide polarity and its aqueous solubility [56,57,58]. The group of Mizumoto (1988) first introduced sulfate groups to monosaccharide structures and found that the sulfated polysaccharide could inhibit T lymphocyte virus, turned sulfated modification into an important direction for structural modification of a polysaccharide. Degree of substitution (DS) is an important indicator to evaluate whether sulfated modification will be successful or not. It is generally acknowledged that the higher degree of substitution proves that the polysaccharide binds more sulfated groups. At present, structure–activity relationship studies mostly focused on the different substitution degrees of sulfated polysaccharides. In 2019, Guo et al. [59] applied sulfur trioxide-pyridine (SO3-Pyr) method to modify *Qingke* β-glucans (THB) and obtained a high degree of substitution of sulfated β-glucans as follows: Ratio of SO3-Pyr to THB of 16.88 g/g, reaction time of 2.03 h, and reaction temperature of 57.54 °C. Results showed that sulfated modification significantly affected the physicochemical structures (water solubilities, apparent viscosities, and molecular weights) and especially the sulfated THB with higher degree of substitution has better solubility and lower molecular weight. Zhang et al. [60] produced the water-soluble fractions (S-TM8-1 to S-TM8-6) with MW from 6.0 to 64.8 × 10^4^ obtained from the sclerotia of *Pleurotus* tuber-regium, with a weight-average molecular mass ranging from 5.76 to 77.4 × 10^4^ (TM8-1 to TM8-6) were sulfated. The modified β-glucans have a more expanded flexible chain in aqueous solution than the native polysaccharides. Sulfonation of the TM8 fractions, of the (1→3)-β-d-glucan, from the sclerotia of *P. tuber-regium* increased the water solubility and chain stiffness. The higher chain stiffness of sulfated fractions may be attributed to the polyelectrolytic effect caused by the sulfate groups substituted on the backbone. 

#### 2.2.2. Carboxymethylation Modification 

Some macromolecular polysaccharides (cellulose, scleroglucan, and pachymaran) are difficult to induce bioactivities because of poor water solubility. Some current reports related to the successful carboxymethylation of cellulose, scleroglucan, and chitin which the water solubility and bioactivities are obviously increased were proved, in comparison with the native polysaccharides that were used in the derivatization. The preparation of carboxymethylated polysaccharides is generally carried out by using chloroacetic acid or sodium mono-chloroacetate as a substrate and reacting with polysaccharides in basic 2-propanone [61,62]. It has been found that carboxymethylation improves the solubility of polysaccharides [63], and also changes the natural conformation of dextran molecules [64,65], together with its physiological activity [35,66]. The structure of the CM glucans in the solution depends upon the degree of substitution and with the increase of the DS, a transition from triple-helical structure through single helical to random structure takes place. Kogan et al. [67] suspended 10 g of the glucan in a mixture of 12.4 mL of aqueous NaOH (300 g/L) and 125 mL of iso-propyl alcohol and stirred the suspension vigorously at 10 °C for 1 h. Subsequently, sodium salt of monochloroacetic acid was added (7.9 g for achievement of the substitution degree 0.5) in 14 mL of water, and the mixture was stirred at 70 °C for 2 h. The excess of NaOH is neutralized with 6 N HCl and the salts removed by dialysis. Finally, the water-soluble yeast polysaccharide derivative—carboxymethylated (1→3)-β-d-glucan (CMG)—a well-known macrophage-stimulator was obtained. Tang et al. [68] prepared and characterized respectively six (1→3)-d-glucan derivatives prepared from yeast cell wall. It indicated that sulfated glucan (S-PJ) had the significant reduction capacity, phosphorylated glucan (P-PJ) had the obvious hydroxyl radical/superoxide anion scavenging activities and anti-lipid peroxidation effect.

#### 2.2.3. Phosphorylation Modification

It has been shown that fructose, glucose, and some other monosaccharides have no natural bioactivities which could be activated after phosphorylated modification. Phosphorylation is a modification method that introduces a phosphate group into a polysaccharide. Common phosphating agents are phosphorus oxychloride, phosphoric anhydride, phosphoric acid or a salt thereof. The reaction of phosphorylation requires strong acid as a catalyst, which often leads to the degradation of the polysaccharide and complex compositions. This material has been demonstrated to possess anti-infective, anti-inflammatory, cardio-protective, and immunomodulating properties [69,70]. Most of the researchers have focused on the artificial synthesis of phosphorylated polysaccharides and some of their analogs. Williams et al. [71] dissolved *Saccharomyces cerevisiae* β-glucan in a mixed system of dimethyl sulfoxide and urea, and added phosphoric acid for phosphorylation to prepare phosphorylated β-glucan. The prepared yeast glucan phosphate has two molecular mass peaks by high-performance size exclusion chromatography, on-line multi-angle laser light dispersion and differential viscosity measurement, and one is MW = 3.57 × 10^6^, accounting for about 2%, the other is MW = 1.10 × 10^5^, accounting for about 98% of the total. Through the Congo red experiment, it was found that the phosphorylated yeast β-glucan can be aggregated into a triple helix configuration in solution. Several studies have shown that both natural and artificially modified phosphorylated polysaccharides have specified medicinal values, due to the fact that the existence of charged phosphate groups can improve water solubility, change molecular weight, and modify chain conformation of polysaccharides. Chen et al. [72] and Huang et al. [73] studied the chain conformation and antitumor activity of phosphate modified *Tuckahoe* β-glucan and *Tuckahoe hyphae* α-glucan, respectively. In 0.15 mol/L NaCl solution, β-glucan phosphate showed a relatively extended compliant chain conformation, and *Tuckahoe hyphae* α-glucan phosphate showed a more extended semi-rigid chain conformation. The introduction of phosphate groups leads to steric hindrance between molecules, increases significantly the solubility of Tuckahoe β-glucan phosphate and *Tuckahoe hyphae* α-glucan phosphate in water and the rigidity of the chain. Compared with glucan that has not been phosphorylated, water-solubility and chain stiffness of the phosphorylated derivative increased, as a result of the introduction of phosphate groups on main chain. β-glucan phosphate and *Tuckahoe hyphae* α-glucan phosphate have significant anti-tumor activity for relatively high molecular masses and the extended chain conformation can increase the interaction between polysaccharide and the immune system, thereby improving anti-tumor activity. Furthermore, selection of derivatives will make the study of their structure–activity relationships more effective and systematic.

### 2.3. Biological Modification

In general, biological modification of polysaccharides mainly refers to enzymatic modification, which is the degradation of β-glucan due to catalysis by enzymes. Since the biological method is mild but environmentally friendly, it is a promising polysaccharide modification technique. Compared with chemical modification, enzymatic modification has the advantages of high specificity, high efficiency and few side effects. By controlling the enzymatic reaction conditions, polysaccharides with different structures can be produced and their physical and chemical properties could also change. It features high security and controllability. 

#### 2.3.1. Enzymatic Modification 

Enzymatic modification has found widespread industrial use in changing the functional properties of β-glucan both in vivo and in vitro. Depolymerization is the most popular catalytic reaction, which depends on the specific enzyme or enzyme mixture employed and on the particular industrial requirement. In some instances, the aim may be to maintain or, at best, only slightly alter the molecular size of functional polysaccharides and produce specific oligosaccharide fragments from them [74,75,76]. The main course of enzymatic degradation is to degrade the backbone of a polysaccharide, before reducing the molecular weight and decreasing its viscosity. The research of Kery et al. [77] showed that β-1,3-glucanase has a highly specific action and acts at the β-l,3-glycosidic bond to randomly cleave β-glucan. The β-1,3-glycosidic bond reduces the molecular weight of β-glucan, causing the β-glucan to degrade into dextrin or oligosaccharide. Duan et al. [78] showed that the yeast β-glucan was hydrolyzed by the production of β-1,3-glucanase by *Trichoderma strain*, and the components after enzymatic hydrolysis included large molecular weight polysaccharides and oligomerization products. Sugar, small molecule oligosaccharides and monosaccharides were the components after enzymatic hydrolysis with MWs > 30 kDa, and accounted for 66.2% of them. The maximum yield of water soluble β-1,3-glucan was 52% when 20% of substrate with 4 U/mL enzyme was incubated at 55 °C in pH 4.5 buffer for 2 h. Yu et al. [79] used yeast glucan as the raw material to obtain a water-soluble glucan by alkaline protease treatment and obtained water-soluble glucan by hydrolysis. Solubilized yeast glucan has good antioxidant activity and inhibitory activity against angiotensin converting enzyme. The enzymatic modification is promising since it can effectively promote the solubility of polysaccharides without chemical residues or environmental pollution. The use of enzymatic modification is currently limited to only a few kinds of polysaccharides. Research and development of other types of enzymes, such as transferase and synthase, will enrich the application of enzyme technology of structural modification of polysaccharide.

#### 2.3.2. Microbial Modification 

Microbial modification refers to the change for the composition and structure of the polysaccharide by the fermentation of microorganisms, thereby changing the physical properties and biological activity of the polysaccharide. Liang et al. [80] developed an efficient and cost-effective microbial fermentation method for the direct production of water-soluble β-1,3-glucan (w-glucan), a coupled fermentation system of *Agrobacterium* sp. and *Trichoderma harzianum* (CFS-AT). W-glucan production reached 17.31 g/ L, with a degree of polymerization of 19–25. It proved that the addition of 10 g·L^−1^ Tween-80 to the CFS-AT enhanced w-glucan production, presumably by loosening the curdlan ultrastructure and increasing the efficiency of curdlan hydrolysis. 

## 3. Advances in Research on Biological Activities of Yeast β-Glucan

Stimulation of the immune system is one of the most important biological functions of β-glucan. Inhibiting the growth and metastasis of cancer cells, preventing bacterial infection, lowering blood cholesterol levels and healing wounds are functions with reported biological activity [81,82,83,84]. Bacha et al. [85] isolated β-glucan from yeast and found that β-glucan (MP, 125 °C) had high stability and antioxidant activity (Tac value (total antioxidant capacity) 0.240 ± 0.0021 g/mg, H_2_O_2_ clearance rate 38%, with a good bile acid binding rate of 40.46%. At the same time, it was found that cholesterol binding in the body and its content in the skin were also reduced. Yeast β-glucan is effective against some markers of inflammation and oxidative stress. Physical and chemical properties of β-glucans are important to their bioactivities. The different structural characteristics of various β-glucans may be the key to their biological activity. β-glucan has not been widely used because of its poor water solubility. Molecular modifications of polysaccharides solve this problem by increasing water solubility. In addition, most of the reported inhibitory effects of (1→3)-β-glucan on endotoxin were found when using soluble (1→3)-β-glucans. However, the establishment of structure–function relationships is possible only if purified and characterized β-glucan is available and selective structural modifications performed. Thus, researchers have modified the structures and properties of natural β-glucan based on structure–activity relationships and have obtained better functionally improved β-glucan. 

### 3.1. Immunomodulation and Anti-Tumor Activity

Both soluble macromolecules and insoluble microparticles of beta-glucan can be used as immunomodulators in innate and acquired immunity. Phosphorylation modification is the most common method to increase antitumor activity and modified polysaccharides have been shown to have anti-HIV activity as a result of sulfated modifications [86,87]. As an immunomodulator, yeast β-glucan is a compound capable of activating immune cells to secrete cytokines and can participate in host-specific and non-specific immunity to improve immune functions (Figure 5) [88,89,90]. They can act directly or indirectly act on the immune system, triggering a series of reactions by activating monocytes, macrophages, and neutrophils. The underlying anti-tumor mechanism of β-glucan may be related to its ability to stimulate the immune system to induce cell apoptosis through a caspase 3-dependent signaling pathway, inhibit cell proliferation possibly via a p53-dependent signaling pathway, and by suppressing VEGF expression to inhibit angiogenesis resulting in slow development of tumors [91,92,93]. The immunomodulatory and antitumor activities of β-glucan depend on its solubility, molecular weight, the degree of branching and conformation changes, and even the difference in the extraction solvent used [94,95,96]. β-(1,3)-glucan has stronger immunostimulatory activity than β-(1,4)-glucan and β-(1,6)-glucan. Modification of insoluble β-glucan by carboxymethylation or sulfonation or physical methods such as ultrasonic depolymerization and mechanical activation can increase the solubility of β-glucan and thereby enhance its biological activity [97,98,99]. Shi et al. [100] mechanically modified *Saccharomyces cerevisiae* β-glucan to increase the dissolution rate from 11% to 37%, and the β-glucan thus prepared had higher immunological activity. Li et al. [101] studied the effects of (1→3)-β-d-glucan on the immunity of S180 tumor-bearing mice and tumor-bearing hosts in *Saccharomyces cerevisiae*. It was demonstrated that (1→3)-β-d-glucan exerted an anti-tumor effect in normal mouse cells without producing any signs of toxicity. After treatment with (1→3)-β-d-glucan, the volume and the weight of S180 tumors decreased significantly. Furthermore, treatment with polysaccharides also showed a dose-dependent increase in the tumor inhibition rate. The results indicated that (1→3)-β-d-glucan could enhance the host’s immune function during suppression of S180 tumor cells treated with (1→3)-β-d-glucan; the cells also showed significant apoptosis characteristics. (1→3)-β-d-glucan increased the ratio of Bax to Bcl-2 at the translational level by up-regulating Bax expression and down-regulating Bcl-2 expression, resulting in the initiation of apoptosis of S180 tumor cells. Taken together, these results indicate that the anti-tumor effect exerted by (1→3)-β-d-glucan may be attributed to the immunostimulatory properties of the polysaccharide and the induction of apoptosis. Further research on these properties and their associated mechanisms will help develop effective anti-tumor agents based on polysaccharides. Qi et al. [102] revealed the importance of different formulations of β-glucan in adjuvant therapy. Yeast β-glucan-activated dendritic cells (DCs) and macrophages were demonstrated to pass the *C*-type lectin receptor dectin-1 pathway. β-glucan particle-activated DC promoted the initiation and differentiation of Th1 and cytotoxic T lymphocytes in vitro. Treatment with oral yeast-derived β-glucan particles elicited an effective anti-tumor immune response and significantly downregulated immunosuppressive cells, resulting in delayed tumor progression. The lack of the dectin-1 receptor completely abolished the anti-tumor effect mediated by particulate β-glucan. In contrast, yeast-derived soluble β-glucans bind to DCs and macrophages, but not to the dectin-1 receptor and do not activate DCs. 

### 3.2. Antioxidant Promotes Wound Healing and Irradiation Resistance

Superoxide dismutase (SOD) is a free radical scavenging enzyme present in the body that plays an active role as an antioxidants. Water-soluble yeast glucan can increase the activity of SOD in serum, an action positively correlated with the dose [103,104,105]. Yeast β-glucan has a strong anti-oxidation effect, which can improve antioxidant activity by removing free radicals, preventing the activity of antioxidants and reactive oxygen species, and improving the antioxidant capacity of the body [106,107,108]. In addition to the above modifications, the modification by acetyl causes great effects on the scavenging action for hydroxyl radicals. Proper acetylation can effectively reduce the hydrogen bonds and activate hydrogen atoms on anomeric carbon. Acetylating yeast β-glucan results in higher antioxidant abilities, such as ferric-reducing power and lipid peroxidation inhibition activity, compared with the untreated [18]. Sener et al. [109] studied the anti-oxidative effect of β-glucan. The results of experiments on mice showed that β-glucan can effectively eliminate chronic oxidative damage in mouse kidney and bladder caused by nicotine, and can also eliminate pus. In vitro cell culture experiments have shown that β-glucan can resist the loss of antioxidant molecules in skin cells and promote keratinogenesis and cell proliferation, while β-glucan can be used as an osmotic inhibitor to eliminate ionic toxicity [11,110]. Some natural polysaccharide glycosaminoglycan activity during wound healing has been demonstrated. Dose-related (1→3)-β-glucan is a potential natural biological response modifier for wound healing and could enhance ulcer healing and increase epithelial hyperplasia, as well as increased inflammatory cell activity, angiogenesis, and fibroblast proliferation [111,112,113]. Water-soluble yeast β-1,3-glucan has excellent radical scavenging effects and can participate in the body’s natural defense mechanisms and accelerate the process of skin wound healing. Ho et al. [110] found that 20~100 mg/mL yeast dextran smear on the injured leg of an animal could significantly accelerate wound healing. Furthermore, the use of 0.1 to 0.2 mg/kg body weight of soluble glucan reduced the infection rate after abdominal surgery. In addition, Harangozó et al. [114] studied the penetration of Cs^+^, Cd^2+^, and Co^2+^ ions in an animal model of human skin (five-day-old rat skin) and found that β-glucan and carboxymethyl-chitosan-glucan reduced the potential toxicological effect of these metals in humans as permeation inhibitors. The dose-dependency of this effect was also demonstrated. Ultraviolet radiation can decrease the number of immune cells in the skin as well as their activity. β-glucan is able to remove free radicals from the body effectively so that macrophages can be protected from free radicals during and after irradiation, so they can function normally [9,81]. Cell culture experiments have shown that carboxymethyl β-glucan can inhibit the consumption of antioxidant molecules under ultraviolet irradiation, promote the growth of keratin, and thus protect skin cells [115]. Studies carried out by Pathchen et al. [116] have shown that water-soluble dextran enhances rapid healing after radiation induced by cobalt-60. Treatment of irradiated mice with β-glucan enhanced the ability of mice to recover and increased bone marrow viability, the white blood cell and spleen cell count. In summary, most of the molecular modification methods, which include chemical, physical, and enzymatic modifications, could increase the antioxidant activity of β-glucan. Among these, sulfated and acetylated modifications are the most commonly used.

### 3.3. Lowering Cholesterol and Blood Sugar

Animals generally do not synthesize β-glucanase and ingested β-glucan cannot be decomposed. Only passive objects in the form of nutrient fibers can absorb these fibers to have the functional effects for example in lowering blood fat and cholesterol levels [117,118,119]. The special structure of yeast β-glucan can promote the release of lipoprotein and fatty acids, and break down the lipids of macromolecules in the blood into small molecules, thus clarifying the serum turbidity caused by hyperlipidemia and significantly lowering blood cholesterol [120,121,122]. Polysaccharides with low molecular weight can easily pass through multiple cell membrane barriers and show better bioactivity. Nicolosi et al. [123] studied 15 obese high cholesterol men, who maintained the same volume during the study and significantly reduced blood total cholesterol by 8% at week 7. After 8 weeks, it decreased by 6%. It is evident that β-1,3-glucan can significantly reduce plasma cholesterol. Kalara et al. [124] used different levels of soluble β-glucan fed to rats to compare their hypocholesterolemic effects. The results showed cholesterol lowering in the serum and liver of the rats. It also proved that the viscous property of soluble β-glucan might reduce absorption or reabsorption of lipids. Peng et al. (2016) reported that β-glucan can adsorb cholesterol and bile acids through active groups on the surface to reduce the reabsorption of these substances in the liver, thereby reducing the cholesterol content [125].

### 3.4. Other Biological Activity

Yeast β-glucan also has the functions of adsorption, elimination of mycotoxins, anticoagulation, and improvement of intestinal functions. Experiments have shown that the sulfate of β-glucan exhibited significant anticoagulant activity, which was strongly influenced by the degree of sulfation, molecular weight and basic polysaccharide structure. The sulfated linear (1→3)-β-d-glucans with DS greater than 1.0 and with a molecular weight between 18 kDa and 50 kDa were shown to be suitable as a potential heparin alternatives. Wang et al. [126] and other studies on the structure–activity relationship of β-glucan sulfate, its molecular weight and degree of sulfation found that the decisive factor of β-glucan anticoagulation is the basic structure of polysaccharide. The action mode of β-glucan is different from that of heparin. It relies on its unique structure to regulate the anti-thrombin AT-III, inhibits the activity of coagulation factor IIa and Xa, and achieves specific stages that interfere with the coagulation process. Fruhauf et al. (2012) reported that yeast β-glucan can adsorb a variety of mycotoxins including zearalenone and aflatoxin [127]. Xu et al. (2015) and Zhang et al. (2006) reported that the unique three-dimensional structure of yeast β-glucan has a good adsorption effect on mycotoxins [128,129]. β-glucans supplementation could improve intestinal health when challenged with *C. perfringens*-induced necrotic enteritis, and inhibit the growth of *Lactobacillus*, *Bifidobacterium,* and *Coliforms*. Besides *L. hyperborean,* LH β-glucan has the most profound reducing effect on coliform counts when compared with the diets supplemented with *L. digitata* and *S. cerevisiae* β-glucans [130,131,132]. It can promote the proliferation of Gram-positive beneficial bacteria in the intestine and inhibit the proliferation of Gram-negative saprophytic bacteria [132]. Improving gastrointestinal function and regulating intestinal flora balance through increasing the secretion of secretory *Ig A* on the intestinal surface [133,134]. In recent years, investigators at home and abroad have carried out much research on the effects of yeast β-glucan on intestinal microbes. As early as 2009, Zhou et al. reported that β-glucan could increase the width, length and mucosal thickness of the rumen nipple of the weaned calf and increase the height of yak small intestine villi and its depth to crypt [135]. In 2010, Wang et al. reported that β-glucan could significantly reduce the number of cecal *E. coli* and *Salmonella* in broilers, and effectively prevent the increase of broiler mortality caused by *Salmonella* infection [136]. Refstie et al. [137] tested the effects of feeding high-purity β-1,3/1,6-glucan on the growth of artificially fed Atlantic salmon and found that the yeast was infected with yeast β-glucan in the diet. The numbers were reduced but it lead to an improved feed efficiency ratio, faster growth, increased protein retention, less diarrhea, and most notably the elimination of induced enteritis. In 2018, Zhang et al. reported that β-glucan increased the number of cecal lactic acid bacteria and bifidobacteria in weaned piglets [138]. Cui et al. believe that β-glucan can inhibit the proliferation of intestinal *E. coli* and has little effect on the proliferation of *Bifidobacteria* or *Lactobacilli*, that is, β-glucan improves intestinal bacteria without affecting the growth of the beneficial bacteria group structure [139].

## 4. Conclusions

β-d-Glucan have been present in the human diet from the beginning of time. Only within the last few years, their positive impact on human and animal health have been elucidated. In many cases, their practical use is limited because of their weak solubility under neutral conditions and their unsuitable hydrophilic/hydrophobic balance. To improve β-d-glucan solubility, several derivatization and molecular weight lowering procedures have being used which are summarized in this review. But there are still some problems with these methods, for instance, the high conversion yields of chemical solubilization modification. However, its main drawbacks are the toxicity of the chemical reagents used and the lack of selectivity. In addition, the introduction of a new group gives the dextran some extra activities, but it changes its original structure and limits its application. This review has summarized the major modification methods of β-glucan, the effect of molecular modifications on β-glucans’ physicochemical properties and the impact of molecular modifications on bioactivity, based on the current pertinent literature. Different molecular modification methods can be applied to obtain a variety of bioactive polysaccharide derivatives. With the continuous exploration of new technologies and new methods, as well as the continuous discovery of new natural polysaccharides, the solubilization and modification of polysaccharides will present new opportunities as well as challenges. In the continuous improvement of polysaccharide molecular modification methods, research on polysaccharides and their derivatives will be further deepened, and the structure–activity relationship of polysaccharides will further clarify that polysaccharides will play a larger role. 

With the high selectivity and substrate specificity of enzymatic solubilization, not only water-soluble β-glucan of a certain molecular weight can be obtained by controlling the degree of hydrolysis, but also the problems of pollution and safety caused by chemical methods can be avoided. Ultrasonic waves can loosen the structure of a polysaccharide, increasing the contact area of the polysaccharide particles with the reagent leading to a high yield of the water-soluble yeast β-glucan. In brief, ultrasound-assisted enzymatic modification is an efficient and controllable way of producing water-soluble β-glucan and should enlarge the field of their potential applications. Different structure types and various bioactivities of polysaccharide derivatives can be obtained through molecular modification, which lays a solid foundation for the analysis of the relationship between polysaccharide structure and function. 

## Figures and Tables

**Figure 1 molecules-25-00057-f001:**
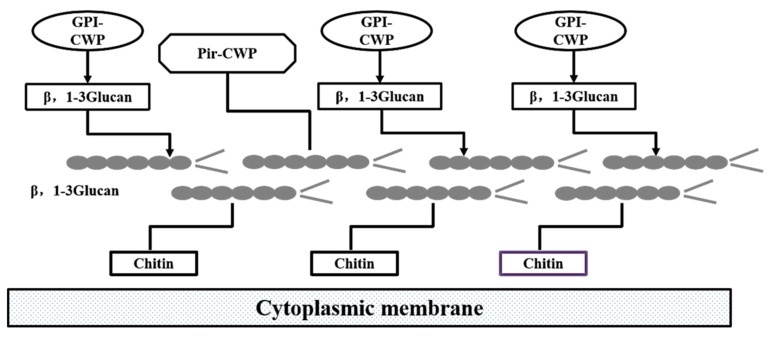
Composition of the yeast cell wall.

**Figure 2 molecules-25-00057-f002:**
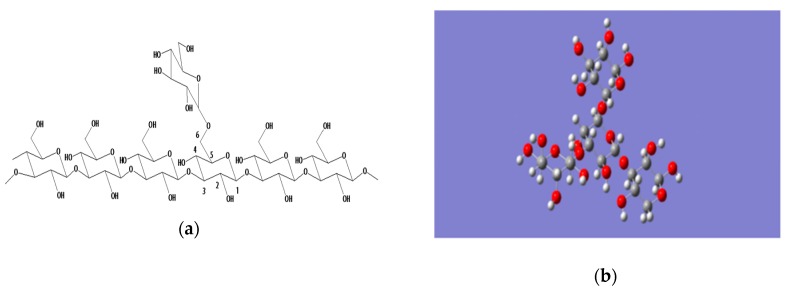
The structure of yeast β-glucan (**a**) One-dimensional structures of yeast β-glucan; (**b**) Three-dimensional structure of yeast β-glucan.

**Figure 3 molecules-25-00057-f003:**
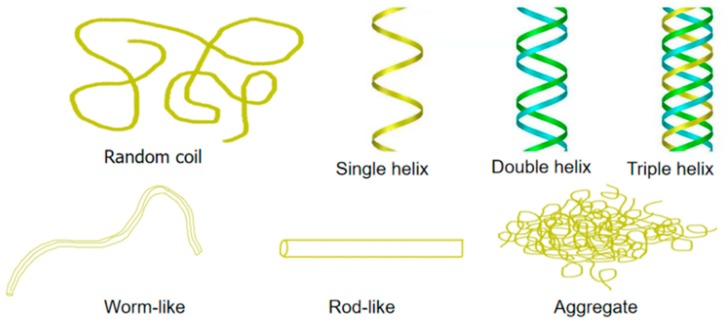
Possible conformations of polysaccharides in aqueous solution.

**Figure 4 molecules-25-00057-f004:**
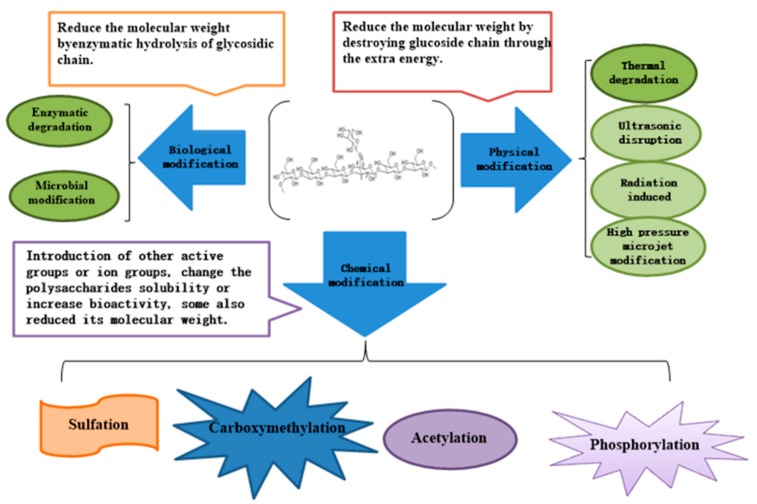
Major methods of molecular modification of β-glucan.

**Figure 5 molecules-25-00057-f005:**
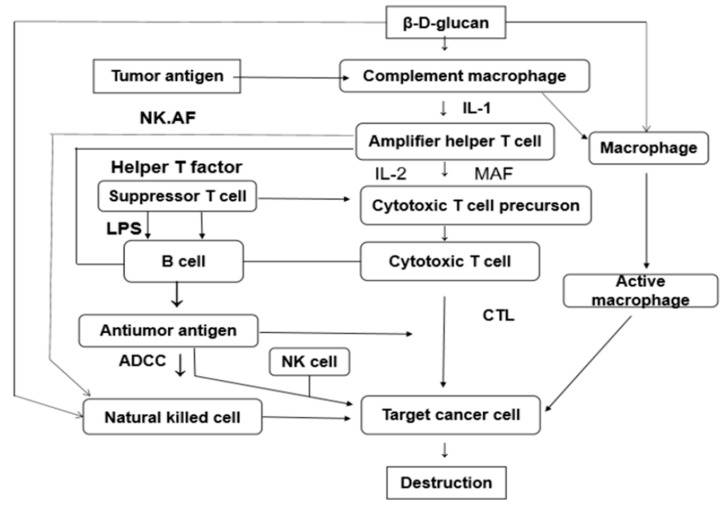
Possible immune mechanism of the β-d-glucans biological response modifier.

**Table 1 molecules-25-00057-t001:** Isolation and purification, water-soluble β-glucan yield and MWs produced by different modification methods.

Water Soluble β-Glucan	Modification Methods	Isolation and Purification	Yield	MW	Ref.
1,3-/1,6-glucan	Heat degradation	Centrifuged at 3,000 rpm for 20 min; filtered through a 0.45 µm disposable syringe filter	89.8%	70.8 × 10^4^ −0.13 × 10^4^	[27]
Black yeast β-Glucan	Irradiation 10, 30 and 50 kGy	_	55.76%, 75.81%, 81.72%	6.2 × 10^4^, 3.2 × 10^4^, 2.5 × 10^4^	[29]
*Poria cocos* β-glucan	Ultrasonic treatment	Ultrafiltration using a membrane with a MW cut-off of 10 kDa	PCS90 6.30 mg/mL (Solubility)	4.3 × 10^4^	[39]
Yeast β-glucan	High pressure micro-jet	Ethanol precipitationcentrifugation	79.3%		[47]
*G. lucidum* β-glucan	Sulfation	Ultrafiltration system (Sartorius. Co. SM 17521), using a 10,000 MW cut-off filter	85%	9.3 × 10^3^	[55]
*Saccharomyces cerevisiae* (1→3)-β-d-glucan	Phosphorylation	1 µm pre-filter Pellicon tangential flow dialyzer (Millipore, Bedford, MA)	70%	1.28 × 10^6^ −0.25 × 10^5^, 3.57 × 10^6^ −1.10 × 10^5^, 12.23 × 10^6^ −3.04 × 10^5^	[71]
Yeast β-1,3-glucan	Enzymic	Sephadex G-100	52%	6.380 × 10^8^, 4.785 × 10^7^, 1.206 × 10^6^	[78]

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
