# Peer review of "Effect of the Modifications on the Physicochemical and Biological Properties of β-Glucan—A Critical Review"

_molecules, 2019, doi:10.3390/molecules25010057_

Round 1

Reviewer 1 Report

General comments:
An effort was made on form and drafting.
However, there are still some incomprehensible sentences that still need to be corrected. It should be noted that part 3 is better written than the previous ones. Maybe the author of this part could review the beginning as well?
It also remains to homogenize the titles and subtitles (see hereafter).
An effort must be made again on the abstract which starts on a sentence only possibly understandable if one removes the word "which". line 17: delete "groups". Finally, the 2 closing sentences of the abstract hardly reflect the title.
Moreover, for the title the concept of "Solubilization modifications" is obscure. Are these structural changes to solubilization, the modification of solubility for activity? If it's the last one, the title does not reflect the content and if it's the first one, the title is not clear.

specific comments

line 62: change "any" into "same"

line 69: rewrite the sentence to "Several parameters characterize the natural b-glucan:....."

line 76: rephrase

line 81: erase "directly or indirectly"

line 91: erase "using...."

line 91: impact of what?

line 111: change "to break" into "by breaking"

line 121: erase "and so on"

line 122 and 141: erase "modification"

line 151: made the "13" in superscript.

line 161: write only : "ultrasonication"

line 166: add "the" before "molecules"

line 167: replace "and is" by "making them"

line 176: longer than what?

line 186: erase "modification solubilization" and write "homogeneization

line 206: erase "application"

line 240 replace "causing" by "changing" and erase "to be changed".

line 246: erase modification

line 249: brake the sence: "....activities. Among them...."

line 250:replace "due" by "thanks"

line 256: define "DS" before using it.

line 258-9: chose between "a lot" and "mostly"

line 261:" obtained a high"

line 275 : completely erase the second sentence!

line 278 : completely erase the second sentence.

line 279 : erase "modification"

line 290-91 :The excess of NaOh is neutralized with... and the salts removed by...."

line 293 :"-" after "stimulator"

line 330: erase "method"

line 331: "enzymatic modification"

line 332: completely erase the second sentence.

line 362:change "of" by "for"

line 527: brake the sence: "....time. Only....years, their positive...health have been elucidated"

Reviewer 2 Report

The present version of the manuscript was greatly improved in comparison to the original. Figure 3 was modified to include other possible conformations of polysaccharides in aqueous solution, and a “Conclusion” section was created. Nevertheless, the first sentence of the Abstract, lines 12-14, still requires revision because it seems to be incomplete. Rephrasing is needed.

Reviewer 3 Report

I am satisfied with responses.

Author Response

This manuscript is a resubmission of an earlier submission. The following is a list of the peer review reports and author responses from that submission.

Round 1

Reviewer 1 Report

This review paper summarized the current literature about solubilization modification and biological activities of β-glucan. Overall, the paper is very well organized and presented, and it would be interesting to researchers with β-glucan.

The resolution of the Figures (1, 3 and 4) should be increased.

As reader, I suggest to add the contents of treatment and use of β-glucan using supercritical fluid technology.

Examples. https://www.sciencedirect.com/science/article/abs/pii/S0963996912001378

https://www.foodnavigator.com/Article/2011/07/04/New-extraction-technique-leads-to-enriched-beta-glucan-Study

etc.

Reviewer 2 Report

This is an interesting review that describes physical, chemical and biological (including enzymatic) modifications of yeast beta-glucan, aiming a higher solubility, to increased its utility. The text covers different types of modifications and describes the aims and results of each procedure, but does not describes the isolation and purifications methods, as well as the yields of each procedure.

Furthermore, the biological activities of yeast beta-glucan are also presented. Some of these activities are affected by the modifications introduced in the molecules, but the effects of contaminations with other cell wall components were not approached.

A curious aspect of this Review is that it includes a “Discussion” section (Pages 13-14, lines 509-525) but does not contain a “Conclusion” section. A Discussion was already included in each section, and it seems much more interesting to have a “General Conclusions” section, than a second “Discussion”.

Nevertheless, the text requires a thorough revision, because there are many errors and inaccuracies. These go from the use of space all through the text (such as Page 1, line 36: “dextran(Figure 1)[2,3]”, should be “dextran (Figure 1) (2,3]”), to position of legend to Figure (such as legend to Figure 1 – Page 2, line 47), and spelling (such as “Butt”, line 514, page 13). Rephrasing is necessary in many points.

Reviewer 3 Report

Dear authors,

I apologize, but I stoped my reviewing at the end of page 5.

Your manuscript is not ready for submission.

Over 5 pages I found hunderds of "space-missings". Five sentences which are absolutely not understandable (words repetitions, two ideas in on single sentence, and so on). Scientifically it is similar: glycan have not one side and other sides, but reducing and non-reducing ends. There is a confusion between "organization" (3D intermolecular interactions) and "conformation" with is determined by the intramolecular organization of the atom .

In the abstract, even the first sentence is tendencious, as glucans are used in pharma and food essentially for their texturation and conservation properties, rather then for their biological activity.

Evenmore important: You write a review and your references count several typing errors, when they are not wrong ex: reference 7 has the wrong title!!!)